# ThinkPrune: Pruning Long Chain-of-Thought of LLMs via Reinforcement Learning

**Bairu Hou**[1]    **Yang Zhang**[2]    **Jiabao Ji**[1]    **Yujian Liu**[1]    **Kaizhi Qian**[2]
**Jacob Andreas**[3]    **Shiyu Chang**[1]

[1]**UC Santa Barbara**    [2]**MIT-IBM Watson AI Lab**    [3]**MIT CSAIL**

Reviewed on OpenReview: `https://openreview.net/forum?id=V51gPu1uQD`

## Abstract

We present THINKPRUNE, a simple yet effective method for pruning the thinking length for long-thinking LLMs, which have been found to often produce inefficient and redundant thinking processes. Existing preliminary explorations of reducing thinking length primarily focus on forcing the thinking process to early exit, rather than adapting the LLM to optimize and consolidate the thinking process, and therefore the length-performance tradeoff observed so far is sub-optimal. To fill this gap, THINKPRUNE offers a simple solution that continuously trains the long-thinking LLMs via reinforcement learning (RL) with an added token limit, beyond which any unfinished thoughts and answers will be discarded, resulting in a zero reward. To further preserve model performance, we introduce an iterative length pruning approach, where multiple rounds of RL are conducted, each with an increasingly more stringent token limit. We observed that THINKPRUNE results in a remarkable performance-length tradeoff on the AIME24 dataset, the reasoning length of `DeepSeek-R1-Distill-Qwen-1.5B` can be reduced by half with only 2% drop in performance. We also observed that after pruning, the LLMs can bypass unnecessary steps while keeping the core reasoning process complete. Code is available at `https://github.com/UCSB-NLP-Chang/ThinkPrune`.

## 1 Introduction

Recent advances in large language models (LLMs) have demonstrated the effectiveness of inference-time scaling through reinforcement learning (DeepSeek-AI, 2025; OpenAI, 2024; Liu & Zhang, 2025; Zeng et al., 2025a), where LLMs learn to produce long and sophisticated reasoning behaviors such as self-refection and verification, significantly increasing their performance on a wide range of benchmarks. However, one key challenge of inference-time scaling is the significant number of tokens produced at inference time, leading to high computational and memory overhead. For example, on the MATH500 (Lightman et al., 2023) benchmark, the `DeepSeek-R1-Distill-Qwen-1.5B` model generates solutions with more than 15,000 tokens on average, while many of the questions could have been solved with fewer than 1,000 tokens by regular LLMs. This highlights the issue of *over-thinking*, where many reasoning steps might be redundant or inefficient (Kumar et al., 2025; Chen et al., 2024; Sui et al., 2025).

There are now some preliminary explorations of limiting the generation length via *budget-forcing* (Fu et al., 2024; Muennighoff et al., 2025), where, when reaching a given token limit, the thinking process is forced to early exit, *e.g.*, by appending an end-of-thinking token and producing an answer right away. Figure 1(a) (left) shows an example output of S1 (Muennighoff et al., 2025), a budget-forcing method, under a low budget (2000 thinking tokens), where the thinking process spent 453 tokens just to understand the problem, and therefore could not complete its second round of thought within the budget. Accordingly, a non-trivial performance drop would occur as the budget gets tighter, as shown in Figure 1(b) (blue lines).

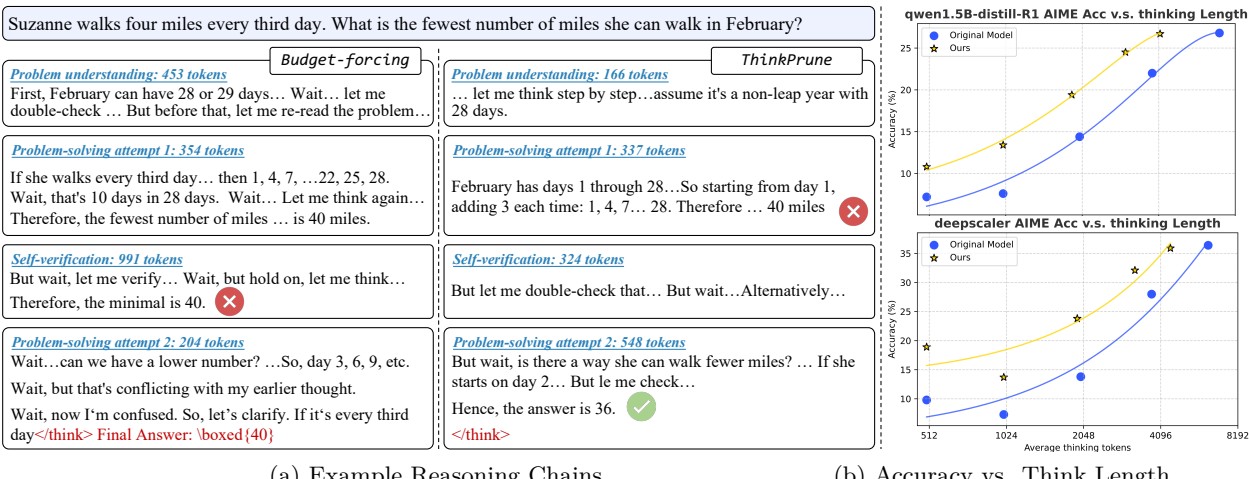

(a) Example Reasoning Chains.    (b) Accuracy vs. Think Length.

Figure 1: Comparison between budget forcing and THINKPRUNE. (a) Example Reasoning Chains: Under a 2000-token thinking budget, applying budget-forcing on the original model uses up all token budgets before identifying the mistake, leading to a wrong answer. In contrast, the model trained with THINKPRUNE solves the problem more efficiently, using fewer tokens and giving the correct answer. (b) Performance of the original LLM and the model after THINKPRUNE training under different thinking token budgets.

However, this is apparently sub-optimal. When the budget is low, a much more sensible solution is to remove the redundant thinking or unimportant steps (such as the problem paraphrasing), rather than maintaining the inefficient thinking and getting it killed. As a result, the performance drop as thinking length shrinks may have been significantly overestimated. So far, there have not been explorations that seek to fine-tune the model to adapt to smaller thinking budgets, and thus, we still do not have a good estimate of how much redundancy there is in the long thinking process that can be removed. Specifically, the following research questions remain unanswered:

- Can we fine-tune an LLM with long CoT to prune its thinking length, while minimizing the performance drop?

- What would be the length-performance tradeoff when the long CoT is pruned?

- What happens to the reasoning chain when it gets pruned? What steps or words are most likely to be pruned?

In this paper, we seek to fill this gap. We propose THINKPRUNE, a simple yet effective length-pruning strategy for LLMs with long CoT, which enforces a maximum generation length during RL training. Specifically, given an LLM with long CoT, we perform continuous RL following the same scheme of DeepSeek-R1, except that we impose a strict token limit during training (*e.g.*, 4,000 tokens for both reasoning and answer tokens). Any tokens beyond this limit are discarded before reward computation. This means that even if the model generates a correct answer beyond the allowed length, it still receives a reward of 0 because the answer is clipped and cannot be extracted. The task performance can be further preserved with an iterative pruning strategy, where multiple rounds of RL are conducted with increasingly more stringent length limits.

Extensive evaluation demonstrates a strong trade-off between generation length and performance for THINKPRUNE. For example, THINKPRUNE reduces the average generation length from 10,355 to 3,574 tokens for the `DeepSeek-R1-Distill-Qwen-1.5B` model, while also improving average accuracy across four math benchmarks. Although there is a slight performance drop of 2% for the `DeepScaleR-1.5B-Preview` and `QwQ-32B` models, our method achieves a comparable reduction in generation length — from 5,914 to 3,370 tokens, and from 8,763 to 4494, respectively. Figure 1(b) (yellow lines) also shows the improved accuracy-thinking length tradeoff. Further analysis shows that THINKPRUNE helps LLMs avoid unnecessary

reasoning steps while maintaining focus on solving the question, as shown in Figure 1(a) (right), where problem understanding drops to 166 tokens. These findings provide valuable insights into improving the inference-time reasoning efficiency for long-COT LLMs.

## 2 Related Work

### 2.1 Reinforcement Learning for LLM Reasoning

Reinforcement learning (RL) has shown strong potential in improving the reasoning abilities of LLMs across various domains, such as math (DeepSeek-AI, 2025) and coding (OpenAI, 2025; Liu & Zhang, 2025), and complicated browser surfing OpenAI (2024). The resulting long-COT LLMs such as OpenAI-o3 (OpenAI, 2024) and DeepSeek-R1 (DeepSeek-AI, 2025) significantly outperform short-COT LLMS and demonstrate that reinforcement learning with verifiable reward (RLVR) can encourage LLMs to develop deep thinking behaviors, such as broad exploration and feasibility checks (Gandhi et al., 2025), without relying on complex reasoning data generation methods like Monte-Carlo Tree Search (Zelikman et al., 2024; Hosseini et al., 2024). However, these behaviors often lead to much longer reasoning traces, sometimes several times longer than those produced by short COT LLMs (Sui et al., 2025; Chen et al., 2025), creating an "overthinking" issue that largely increases inference costs (Kumar et al., 2025). Several recent works have shown that this extended reasoning often includes redundant or unnecessary verification and reflection, even on simple problems (Wang et al., 2025; Ji et al., 2025). Our work follows the standard RL training pipeline without changing the reward function and shows that it is possible to retain strong reasoning performance while significantly reducing overthinking.

### 2.2 Efficient Long Chain-of-Thought LLM

Several works have aimed to improve token efficiency in long chain-of-thought (CoT) LLMs. For example, KimiTeam et al. (2025), Chen et al. (2024), and Shen et al. (2025) reduce the length of reasoning by adding a length penalty during RL training. Other works, such as Hao et al. (2024) and Geiping et al. (2025), represent reasoning as an optimization over latent vectors instead of text tokens, which helps shorten the reasoning process. While these methods are effective when training long CoT LLMs from short COT ones, there have been few works on reducing the reasoning length for trained LLMs. The most relevant works to ours are test-time methods that shorten reasoning with early exit strategies (Muennighoff et al., 2025; Fu et al., 2024; Zeng et al., 2025b). These methods add stopping tokens or limit the maximum number of generated tokens during testing. However, they often lead to significant performance drops, especially when early stopping occurs too soon in the reasoning process. In contrast, our work presents a simple and effective method with further RL training to reduce reasoning length without sacrificing performance.

## 3 Method

### 3.1 Overview

Denote $\mathcal{M}_{\boldsymbol{\theta}_0}(\cdot)$ as an LLM capable of performing long CoT. Given a query $\boldsymbol{q}$, a sample answer, along with a long reasoning chain, is sampled from the LLM's output distribution, $i.e.$, $\boldsymbol{Y} \sim \mathcal{M}_{\boldsymbol{\theta}_0}(\boldsymbol{q})$. Our goal is to fine-tune the LLM, such that its output length is reduced while the overall performance is desirably maintained.

THINKPRUNE is a simple yet effective RL strategy to achieve the length reduction goal. In what follows, we will first introduce the RL framework of THINKPRUNE, and then introduce an iterative length reduction strategy to better maintain the performance.

### 3.2 Reinforcement Learning with Length Clipping

THINKPRUNE adopts a similar RL scheme to DeepSeek-R1 (DeepSeek-AI, 2025) while reducing the generation length. Specifically, we adopt the group relative policy optimization (GRPO) algorithm Shao et al. (2024).

The reward function is almost the same as the DeepSeek-R1 framework, *except* that a *length clipping* is added. Formally, denote $L$ as the length limit. The reward function is defined as follows:

$$R(\boldsymbol{Y}, \boldsymbol{q}; L) = \begin{cases} 1 & \text{if an answer can be extracted from clip}(\boldsymbol{Y}, L) \text{ and is correct,} \\ 0 & \text{otherwise;} \end{cases} \tag{1}$$

where $\text{clip}(\boldsymbol{Y}; L)$ represents clipping the output $\boldsymbol{Y}$ to length $L$. In other words, the only difference from the DeepSeek-R1 reward is that the model-generated output is clipped to $L$ before the reward is evaluated. In this way, output with length above $L$ would not be able to produce a valid answer since it is cut off, thus receiving a zero reward. This clipping operation effectively encourages the model to produce answers below the length limit. Such reward design is very simple, so it does not involve any hyperparameter tuning or reward engineering, and can ensure training stability. Also, since it involves minimal tweaks to DeepSeek-R1's training strategy, the proposed solution helps to maintain the long reasoning capabilities inherent in the base model.

During training, we also append a simple system prompt into each training example to explicitly tell the model the length limit, such as *The output of the assistant should be within L tokens*, to explicitly tell the model the length limit. The full system prompt for each model is shown in Appendix A.1.

### 3.3   Iterative Length Pruning Strategy

The success of the proposed algorithm relies on the choice of the length limit, $L$ — if $L$ is set too stringently compared to the original output length of the base model, then the task performance can be seriously compromised.

Drawing inspiration from the iterative pruning strategy for reducing model parameters (Han et al., 2015), we propose an iterative length pruning scheme. Denote $L^*$ as the target length constraint, we introduce a length schedule, $L_1 > L_2 > \cdots > L^*$. At each iteration $t$, we reduce the length constraint to $L_t$, and further fine-tune the model $\boldsymbol{\theta}_{t-1}$ from last iteration to $\boldsymbol{\theta}_t$, using the RL procedure described in Section 3.2. Such an iterative pruning strategy ensures that the LLM learns to recover the performance by gradually compacting its output reasoning chain.

A critical design choice of the iterative length pruning is the stopping criterion for each iteration of the RL training. In our implementation, we utilize AIME22 and AIME23 as the dataset as a validation set to choose the best checkpoint for the next RL iteration. To better balance model performance and generation length reduction, we allow up to a relative 10% drop in pass@1 accuracy on AIME-22 and AIME-23 compared to the original model. Among the checkpoints that meet this criterion, we select the one with the shortest average output length as the starting checkpoint.

## 4   Experiment

In this section, we conduct empirical evaluations to assess the effectiveness of our proposed method. We first present the experiment setup in Section 4.1. Then, we present the experiment results in Section 4.2.

### 4.1   Experiment Setup

**Backbone models.**   Representative open-sourced long reasoning LLMs include DeepSeek-R1 (DeepSeek-AI, 2025) and QwQ (QwenTeam, 2025), along with their distilled variants. In our experiments, we use three models from these families: Distill-Qwen-1.5B, DeepScaleR-1.5B-Preview, and QwQ-32B.

We group these models into two categories based on the extent of their training: *unsaturated* and *saturated* models. Specifically, while Deepseek-R1-Distill models have been widely used, these models are directly fine-tuned on the output of DeepSeek-R1 in a supervised fine-tuning manner instead of RL, which could limits their full potential. Previous work, such as Luo et al. (2025), has shown that further training these models with RL can improve their performance, especially on math benchmarks. Based on this, we treat Distill-Qwen-1.5B as an unsaturated model and DeepScaleR-1.5B-Preview as its saturated version, since

it is further trained with RL. Similarly, QwQ-32B is also trained with RL and is considered a saturated model. This selection of models, covering both unsaturated and saturated types, allows us to more thoroughly evaluate the effects of thinking length pruning.

**Training datasets.** Previous work (Ye et al., 2025) has shown that even a small but high-quality training dataset can improve the LLM performance via RL. Therefore, we utilize a small number of data for model training by only using the historical AIME and AMC math questions (AMC, 2025). We use the preprocessed data from Prime (Cui et al., 2025) and take the AIME-AMC subset for training. In total, the training dataset consists of 2470 distinct training examples.

**Comparisons.** We compare with the original backbone models without pruning. Additionally, we include the budget-forcing method (Muennighoff et al., 2025) for length reduction, which enforces a maximum number of thinking tokens by appending the end-of-thinking token delimiter (detailed implementation is described in Appendix A.1). For our method, we report the following two variants:

- One-shot length pruning: We set the maximum length to 4,000/3,000/2,000 tokens respectively and directly perform RL to reduce the generation length of the LLM.

- Iterative length pruning: Starting from a higher generation length budget, we perform multi-round RL training and gradually decrease the maximum length after each iteration.

**Implementation details.** We use the Verl (Sheng et al., 2024) RL framework for high-performance training. All models are trained with a batch size of 128. The number of rollouts for each question is set to 16 following prior works (Zeng et al., 2025a; Liu & Zhang, 2025). We use the GROP algorithm (Shao et al., 2024). For both one-shot and iterative pruning, we employ the same checkpoint selection strategy as mentioned in Section 3.3, where the checkpoint with the shortest average output length while maintaining a relative 10% accuracy drop is selected.

**Evaluation configurations.** We follow prior works to include the following evaluation datasets: MATH-500 (Lightman et al., 2023), AIME24 (AMC, 2025), AMC23 (AMC, 2025), and OlympiadBench (He et al., 2024). We use the versions of these dataset hosted in the Qwen2.5-Math GitHub repository for ease of reference. Following DeepSeek R1 (DeepSeek-AI, 2025), we set the maximum generation length (including both the thinking tokens and answer tokens) to 32,768 tokens for all the models, far above the maximum token length during our training. For each testing question, we sample $N$ outputs with a temperature of 0.6 and a top-p value of 0.95, and we report the average accuracy of these $N$ outputs. The number of samplings varies depending on the model size and dataset size. Specifically, for the two 1.5B models, we use $N = 64$ for AIME24 and AMC23 and $N = 16$ for MATH500 and OlympiadBench. This also refers to the evaluation hyper-parameters of DeepSeek R1, where the number of sampled responses are adjusted between 4 and 64 depending on the test set size to balance the variance and evaluation cost. For `QwQ-32B`, we sample $N = 16$ responses for each question, given its large size and high computational cost.

Since the accuracy evaluation requires complex evaluation of mathematical expressions, we adopt the math evaluator from Qwen-2.5-math (DeepSeek-AI, 2025), which provides robust answer extraction and advanced expression comparison.

## 4.2 Main Experiment: Length Pruning

**One-shot length pruning enables significant thinking length reduction.** We first evaluate the effectiveness of one-shot pruning in Table 1 (top 3 rows in each section), where the LLMs are directly trained with a length limit 2k/3k/4k. We highlight the following observations. First, the simple strategy can effectively reduce the generation length with moderate negative effect on the performance. For the unsaturated model, `DeepSeek-R1-Distill-Qwen-1.5B`, the length reduction rate can be up to 50% with one-shot length pruning. At the meantime, the average performance is well-maintained and even slightly improved. For saturated models like `DeepScaleR-1.5B-Preview` and `QwQ-32B`, we observe a 40–50% reduction in token usage, with moderate performance degradation. This highlights the promising efficiency gains through length pruning,

Table 1: Performance visualization of THINKPRUNE. The accuracy is measured by sampling multiple responses from the LLMs and taking the average to reduce variance.

| | Accuracy | | | | | Generation Length | | | | |
|---|---|---|---|---|---|---|---|---|---|---|
| | MATH 500 | AIME | AMC | Olympiad Bench | Avg. | MATH 500 | AIME | AMC | Olympiad Bench | Avg. |
| **DeepSeek-R1-Distill-Qwen-1.5B** | | | | | | | | | | |
| Original Model | 82.9 | 29.4 | 70.3 | 44.7 | 56.8 | 5560 | 15484 | 10030 | 11526 | 10355 |
| THINKPRUNE-4k | 83.8 | 29.0 | 73.6 | 46.5 | 58.2 | 2709 | 8301 | 4388 | 5529 | 5232 |
| THINKPRUNE-3k | 83.7 | 27.8 | 71.8 | 44.9 | 57.1 | 2557 | 7968 | 4096 | 5140 | 4940 |
| THINKPRUNE-2k | 82.9 | 27.0 | 72.2 | 45.6 | 56.9 | 2356 | 7574 | 3755 | 4913 | 4650 |
| THINKPRUNE-4k→3k | 83.9 | 26.9 | 71.4 | 46.0 | 57.1 | 2209 | 6389 | 3422 | 4229 | 4062 |
| THINKPRUNE-4k→3k→2k | 83.2 | 27.1 | 73.2 | 46.2 | 57.4 | 1938 | 5631 | 3039 | 3687 | 3574 |
| **DeepScaleR-1.5B-Preview** | | | | | | | | | | |
| Original Model | 88.5 | 40.3 | 81.2 | 52.7 | 65.7 | 3084 | 9463 | 5202 | 5907 | 5914 |
| THINKPRUNE-4k | 87.1 | 37.2 | 80.2 | 51.4 | 64.0 | 2212 | 6366 | 3516 | 4055 | 4037 |
| THINKPRUNE-3k | 86.5 | 34.3 | 78.8 | 50.6 | 62.6 | 1991 | 5809 | 3122 | 3583 | 3626 |
| THINKPRUNE-2k | 86.1 | 33.3 | 77.7 | 49.7 | 61.7 | 1880 | 5528 | 2961 | 3348 | 3429 |
| THINKPRUNE-4k→3k | 87.1 | 38.4 | 79.9 | 51.6 | 64.2 | 2086 | 5869 | 3278 | 3731 | 3741 |
| THINKPRUNE-4k→3k→2k | 86.9 | 36.5 | 79.4 | 50.1 | 63.2 | 1881 | 5301 | 2963 | 3334 | 3370 |
| **QwQ-32B** | | | | | | | | | | |
| Original Model | 95.1 | 78.8 | 97.5 | 71.1 | 85.6 | 4289 | 13822 | 7442 | 9497 | 8763 |
| THINKPRUNE-4k | 94.0 | 76.3 | 95.8 | 68.2 | 83.5 | 2552 | 8787 | 4173 | 5687 | 5300 |
| THINKPRUNE-3k | 94.0 | 75.0 | 95.8 | 68.6 | 83.3 | 2341 | 8308 | 3943 | 5413 | 5001 |
| THINKPRUNE-2k | 93.5 | 73.3 | 95.5 | 68.6 | 82.7 | 2133 | 8232 | 3770 | 5160 | 4824 |
| THINKPRUNE-4k→3k | 94.0 | 73.8 | 96.1 | 67.6 | 82.9 | 2308 | 8176 | 3959 | 5301 | 4936 |
| THINKPRUNE-4k→3k→2k | 93.8 | 72.5 | 95.9 | 67.3 | 82.4 | 2162 | 7631 | 3441 | 4742 | 4494 |

especially when models are initially over-generating. Second, we observe a consistent tradeoff between length and performance for all models under one-shot pruning. As we lower the token limit from 4k to 2k, the number of tokens goes down and the accuracy drops slightly. This suggests that cutting more reasoning tokens aggressively may also limit the reasoning capabilities of the LLMs.

Another interesting phenomenon is that even though the model is trained with an explicit length limit, it often goes beyond that limit at test time. Particularly, we observe that although trained with an explicit length limit, the model can still generate long responses when the problem becomes more difficult such as on the AIME dataset. This shows that our length pruning does not hurt the model's deep thinking behavior, and it maintains the ability to perform complex reasoning for difficult questions.

**Iterative length pruning benefits performance preservation and length reduction.** The second question we aim to explore is whether iterative length pruning can better main the original LLM performance with a decent thinking length pruning compared to one-shot pruning. As shown in Table 1 (last two rows in each section), we start with LLMs trained using a 4k length limit and then iteratively apply RL with reduced length limits from 3k to 2k. Our findings are as follows. **First**, for the two 1.5B models, iterative pruning leads to better performance with either shorter or similar response lengths compared to one-shot pruning under the same final length limit. For example, on DeepScaleR, THINKPRUNE-4k→3k→2k outperforms the one-shot 2k model by 1.5% in accuracy while using 59 fewer tokens on average. Similarly, pruning from 4k to 3k leads to better results than directly pruning to 3k. **Second**, for the QwQ-32B model, performance drops slightly after length pruning, with an average decrease of 2.7%. Unlike the 1.5B models, iterative pruning does not help recover the lost performance. We believe this is because it struggles to maintain performance when the generation token budget is too tight. We will discuss this in more detail in the next paragraph.

**Performance-length trade-off along the pruning process.** We observe that the experiment results in Table 1 indicates a positive correlation between the reasoning length and model performance on different benchmarks. To better understand the trade-off between reasoning length and model performance, we visualize how these two change during the length pruning training process on the AIME24 dataset (see

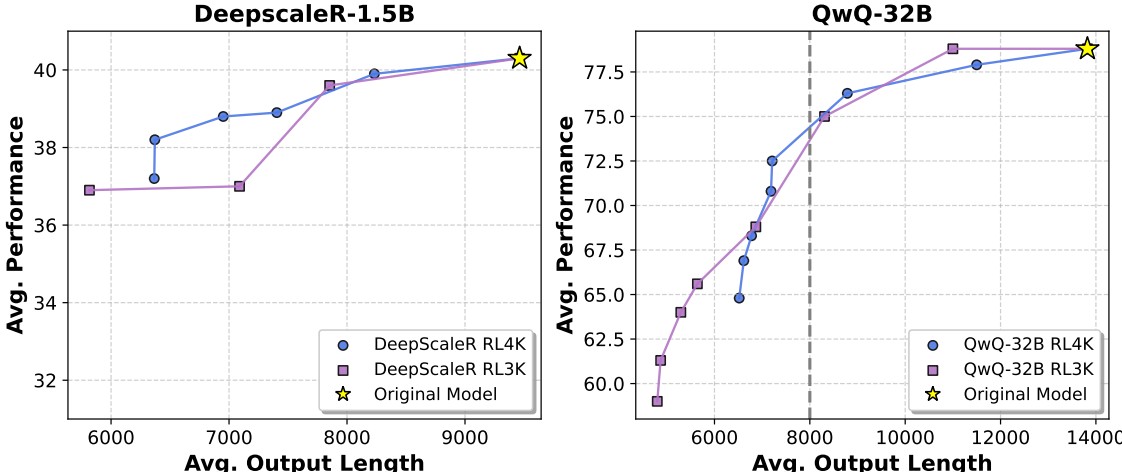

Figure 2: Performance-length trade-off along the length pruning training process evaluated on the AIME24 dataset. The generation length drops quickly in the early stages of training with minor performance drop.

Figure 2). During training, we evaluate the model every 20 steps and record both the performance and the average output length. We then identify the most effective checkpoints (the ones at the frontier of performance-length trade-off), defined as the checkpoints that achieve the best performance among all the checkpoints whose average output length is shorter than themselves. This gives us a clear picture of the best performance the model can reach at each length range, helping us visualize the efficiency boundary of the pruning process.

We highlight the following observations: **First**, mild length pruning leads to strong efficiency gains with minimal performance drop. When reasoning length reduction is moderate, there exists a favorable trade-off: significant token reductions can be achieved with only minor drops in performance. For example, we can reduce the average generation length of `QwQ-32B` from 14K to 8K while maintaining its performance close to the original model. **Second**, for the `QwQ-32B` model, we observe a distinct critical threshold (marked by the vertical dotted line in Figure 2). Beyond this threshold, further reducing the reasoning length causes a sharp and significant drop in performance. We also find enforcing 4k length limit leads to even worse results than 3k length limit on `QwQ-32B`. This drop partially explains why iterative pruning performs worse on `QwQ-32B` than on smaller models like 1.5B. While optimizing hyperparameters and the training data for `QwQ-32B` may lead to better performance, we leave the exploration to future work due to the heavy computation cost.

**Inference-time trade-off with budget-forcing.** We further study how different long-CoT LLMs perform when applying the budget-forcing method (Muennighoff et al., 2025). Detailed budget-forcing prompt can be found in Appendix A.1. Figure 3 shows the trade-off between performance and output length on the Math-500 dataset (left) and the AIME-24 dataset (right), comparing models before and

Table 2: Performance comparison between our method and the length-based rewards.

| | Accuracy | | Length | |
|---|---|---|---|---|
| | KIMI-1.5 | THINKPRUNE | KIMI-1.5 | THINKPRUNE |
| MATH500 | 83.0 | 83.2 | 2108 | 1938 |
| AIME | 27.1 | 27.1 | 6617 | 5631 |
| AMC | 72.5 | 73.2 | 3519 | 3039 |
| OlympiadBench | 44.3 | 46.2 | 4354 | 3687 |

after applying THINKPRUNE with budget-forcing. We highlight two key observations: **First**, THINKPRUNE significantly improves performance under a limited thinking token budget. For example, THINKPRUNE-4k→3k→2k consistently outperforms the original model when using the same number of thinking tokens. On Math-500, it reaches similar accuracy while using only about 50% of the original thinking tokens for `Qwen1.5B-Distill-R1`. This suggests that THINKPRUNE helps the model think more efficiently and make better use of a limited token budget. **Second**, THINKPRUNE reduces more thinking tokens on easier Math-500 problems than on AIME-24. This suggests that redundancy in reasoning is related to problem difficulty and that length pruning can adaptively remove unnecessary thinking for questions at different difficulties.

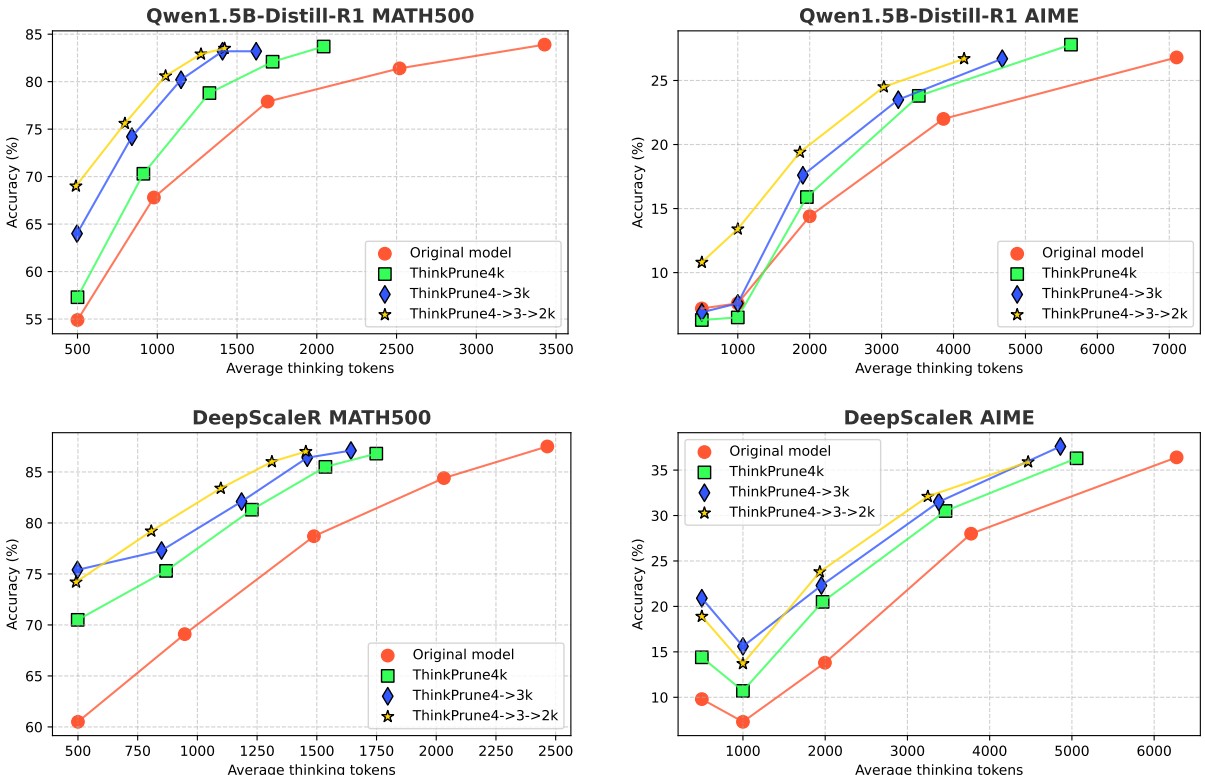

Figure 3: Inference-time thinking length vs. performance trade-off for different long-CoT LLMs with THINKPRUNE after applying budget-forcing.

**Comparison with length-based rewards.** We also compare our method with existing approaches that incorporate length-based rewards to reduce reasoning length. Specifically, we adopt the length reward introduced in Kimi-1.5 (KimiTeam et al., 2025), which adds a length-based penalty to overlong responses. We compare this method with our iterative pruning method, THINKPRUNE-4k→ 3k→ 2k. The backbone model is DeepSeek-R1-Distill-Qwen-1.5B. The data mixture and hyper-parameters used for the baseline training are the same as our method. The experiment results are shown in Table 2. Our method achieves both higher accuracy shorter output length compared to the Kimi-1.5-style length-based reward baseline. This demonstrates that our approach is not only simpler but also more effective. Additionally, we perform comparisons with more baselines on more backbone models in Appendix A.3, which consistently demonstrate that our method provides better length-accuracy trade-off compare to baseline methods.

**Cross-dataset evaluation.** Additionally, to further verify the generalization ability of out method beyond math reasoning tasks, we evaluate our method on GPQA, MMLU, and MMLU-Pro, which cover broader domains such as science and general knowledge. The backbone model is DeepSeek-R1-Distill-Qwen-1.5B trained only on math question. The results are shown in Table 3.

We highlight the following key observations. First, THINKPRUNE can naturally generalize to non-math domains when only trained exclusively on math reasoning data. On all benchmarks, there is only minimal performance change (within ±1.5%) compared to the original models. Second, the reasoning length is significantly reduced across all benchmarks. For example, THINKPRUNE-4k→ 3k→ 2k shortens the reasoning chain by 55–65% (*e.g.*, from 10,355 to 3,574 tokens on Math, and from 10,072 to 4,639 on GPQA), while maintaining nearly identical accuracy. These results further confirm the generalization ability of our method, which can reduces redundancy while maintains the performance even on unseen domains.

Table 3: The accuracy and generation length of THINKPRUNE across different domains.

| | Accuracy | | | | Generation Length | | | |
|---|---|---|---|---|---|---|---|---|
| | Math | GPQA | MMLU | MMLU-Pro | Math | GPQA | MMLU | MMLU-Pro |
| Original Model | 56.8 | 32.7 | 49.1 | 39.4 | 10355 | 10072 | 2816 | 5661 |
| THINKPRUNE-4k | 58.2 | 34.5 | 48.8 | 39.9 | 5232 | 6465 | 1685 | 3515 |
| THINKPRUNE-4k→ 3k | 57.1 | 35.6 | 48.8 | 39.4 | 4062 | 5349 | 1395 | 2982 |
| THINKPRUNE-4k→ 3k→ 2k | 57.4 | 34.2 | 48.3 | 39.7 | 3574 | 4639 | 1243 | 2687 |

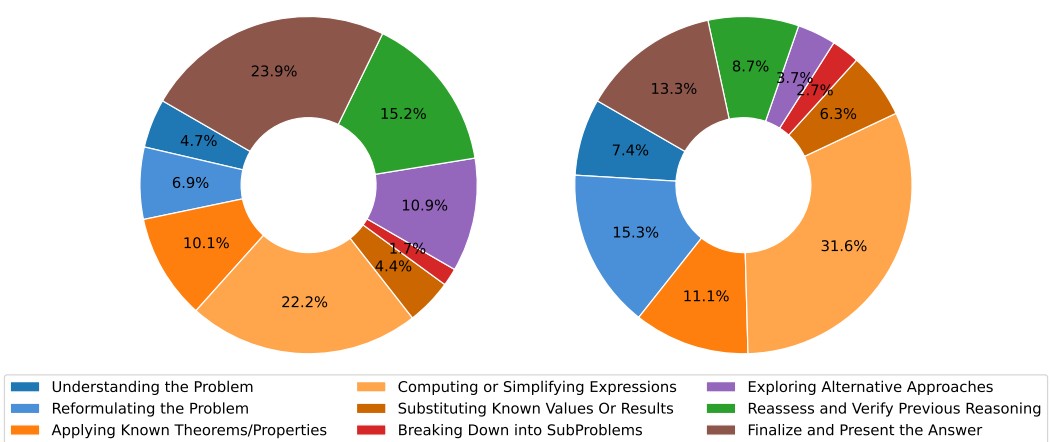

Figure 5: Reasoning behavior change before (left) and after (right) length pruning.

## 4.3 Reasoning Behavior Analaysis

In this section, we study the reasoning behavior of long-COT LLMs after length pruning. We focus on answering the following two questions: 1. How does the reasoning behavior of LLMs change, and what gets removed most after pruning? 2. How does the pruning affect the readability of model-generated reasoning?

**Reasoning-related keyword frequency change.** Figure 4 illustrates the frequency of specific reasoning-related keywords per response in the MATH500 dataset using the `DeepSeek-R1-Distill-Qwen-1.5B` model, both before and after applying our pruning method. Specifically, we count the number of occurrences of each keywords within the responses of LLMs and then normalize the count by the number of tokens in the responses, which gives us the frequency of each word in 1000 tokens.

The figure is divided into three sections. The left section contains phrases that signal hesitations or self-corrections, which we find undergo a significant drop in frequency after pruning. This indicates that the model would hesitate less. The middle section

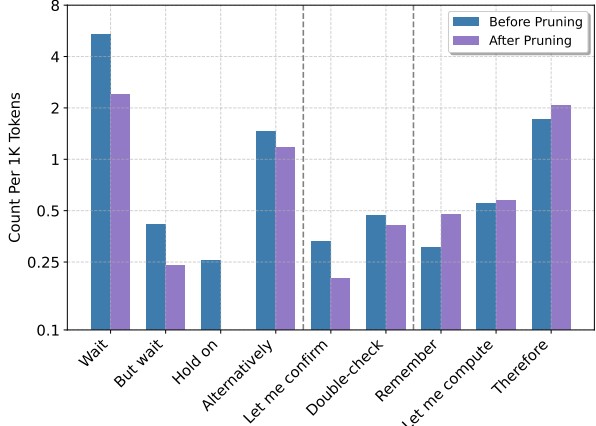

Figure 4: Reasoning-related keyword frequency change before and after length pruning.

contains phrases that signal self-verification, which undergo a slight drop in frequency after pruning. This indicates that the model would sometimes skip the self-verification steps, which may sacrifice a little bit of accuracy, to save the token length. Finally, the right section contains phrases that signal core computation and reasoning, which have a slight increase in frequency despite the overall reduction in length. This indicates that the model would maintain the reasoning process, because that is the core process that contributes to the final solution.

**Reasoning behavior change.** We further analyze the change in reasoning behaviors that the model utilizes to solve a problem. Specifically, we first prompt GPT-4o to summarize 9 frequent problem-solving phases, such as 'understanding the problem' and 'applying known theorems/properties' (complete list in Appendix A.2). Then, for each model-generated solution, we prompt GPT-4o to split the output into chunks and label each chunk with one of these phases. We measure the number of reasoning steps in each phase by counting segments separated by double newlines ("\n \n"). Figure 5 shows the distribution of these reasoning steps before and after pruning. As can be observed, the model would spend less time on relatively redundant steps, such as 'finalize and present the answer', 'reassess solution', and 'explore alternative approaches'. Meanwhile, we observe an increase in the percentage of core problem-solving steps, including 'applying known theorems/properties' and 'computing or simplifying expressions'. These observations further confirm that the model would focus more on the core problem-solving steps and save on peripheral steps.

Table 4: Reasoning trace perplexity on different models.

|  | PPL | Avg. Acc |
|---|---|---|
| **DeepSeek-R1-1.5B-Distill** | | |
| Original Model | 1.91 | 82.9 |
| THINKPRUNE | 1.90 | 83.2 |
| **DeepScaleR-1.5B-Preview** | | |
| Original Model | 1.95 | 88.5 |
| THINKPRUNE | 2.02 | 86.9 |
| **QwQ-32B** | | |
| Original Model | 2.37 | 95.1 |
| THINKPRUNE | 2.24 | 93.8 |

**Readability evaluation.** One common concern in RLVR is that the heavy RL training may reduce the readability of the model's reasoning, resulting in mixed language or non-readable reasoning trace as shown in R1-Zero model (DeepSeek-AI, 2025). To examine whether THINKPRUNE would also introduce similar issues, we measure the perplexity of generated reasoning traces on the Math-500 dataset for original model and THINKPRUNE-4k→ 3k→ 2k using Qwen2.5-Math-7B. As shown in Table 4, our pruning method does not significantly affect reasoning readability — the perplexity remains nearly identical to the original model. Figure 6 in Appendix A.4 shows an example reasoning trace from the Distill-R1-1.5B LLM before and after pruning. As shown in the example, the original LLM repeatedly checks its previous reasoning multiple times, leading to heavy, redundant reasoning steps. On the contrary, the reasoning trace remains fully readable and focused on solving the problem after pruning, with more efficient problem-solving steps and only one self-verification step.

## 5 Conclusion

In this paper, we proposed THINKPRUNE to reduce the reasoning length of long CoT LLMs. THINKPRUNE introduces a length constraint during RL training, which discards unfinished thoughts and answers when sampling responses. To maintain model performance, we apply an iterative pruning strategy that gradually tightens the length limit over multiple rounds training. Experiments show that THINKPRUNE reduces the reasoning length and achieves a strong performance-length trade-off. Further analysis shows that THINKPRUNE effectively removes redundant steps while preserving key reasoning processes.

## 6 Acknowledgement

The work of Bairu Hou, Jiabao Ji, Yujian Liu, and Shiyu Chang was partially supported by National Science Foundation (NSF) Grant IIS-2338252, NSF Grant IIS-2207052, NSF Grant IIS-2302730, and the Open Philanthropy Research Award. Jacob Andreas was supported by NSF grants IIS-2212310 and IIS-2331117. This work was additionally supported by the MIT–IBM Watson AI Lab.

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

# A Appendix

## A.1 Implementation Details

**System prompt used for training.** The system prompt for DeepSeek-R1-Distill-Qwen-1.5B and Deep-ScaleR during training is shown in Table 5 to align with the original RL training of DeepSeek-R1. For QwQ-32B, we use a much similar prompt, "you are a helpful assistant. Your output should be within {N} tokens.".

Table 5: Template for DeepSeek-R1-Distill-Qwen-1.5B and DeepScaleR. {N} will be replaced with the length limit for training (*e.g.*, 2000 and 4000).

---

A conversation between User and Assistant. The user asks a question, and the Assistant solves it. The assistant first thinks about the reasoning process in the mind and then provides the user with the answer. The reasoning process and answer are enclosed within <think> </think> and <answer> </answer> tags, respectively, i.e., <think> reasoning process here </think> <answer> answer here </answer>. The output of the assistant should be within {N} tokens.

---

**Implementation of budget forcing.** We follow the official implementation of budget forcing in S1 Muennighoff et al. (2025) and made minor changes. Since the original implementation is coupling with the `lm-harness-eval` framework (Gao et al., 2024), we revise the code to remove such dependency. Also, to stop the thinking process of the LLM, we append "</think>\n\n**Final Answer:**\n\n" to the generation of the LLM instead of "< |im_start| >answer\nFinal Answer:" used by the S1 model. This is because we empirically find the DeepSeek-R1-Distill-Qwen-1.5B model typically summarize its final answer starting with "**Final Answer:**\n\n". Finally, we sample multiple responses to reduce the variance in model performance instead of using greedy decoding as the original implementation.

## A.2 Analyze the Reasoning Behavior Change

We use GPT-4o to analyze the reasoning behavior of LLMs by segmenting their long-form solutions into high-level problem-solving phases. The prompt used for this task is shown in Figure 7. Since the model-generated solutions are very long, we only require GPT-4o to output of the first step and last step in each chunk to represent that chunk. Since the model-generated solutions are often very long, we ask GPT-4o to return only the first and last reasoning steps of each phase, which serve as markers to define the boundaries of each chunk.

To align these chunks with the original model output, we use string matching to locate the start and end positions of each chunk in the raw text. Within each matched chunk, we estimate the number of reasoning steps by counting the number of "\n \n" delimiters. This gives us a step-level breakdown of how much reasoning is devoted to each phase.

## A.3 Additional Comparison with Length-based Rewards

In this section, we present additional comparisons with length-based rewards. Besides length reward introduced in Kimi-1.5 (KimiTeam et al., 2025), we also include EFFICIENTREASON (Arora & Zanette, 2025) for comparison, which reduce the length via a group-relative length penalty with a tunable coefficient to control the strength of the penalty. We further extended our evaluation to a larger backbone model, DeepSeek-R1-Distill-Qwen-7B, to provide a more comprehensive evaluation. The training recipe and dataset for the 7B model are exactly the same as DeepSeek-R1-Distill-Qwen-1.5B. The experiment results are in Table 6. We group models with similar accuracy to better demonstrate the accuracy-length tradeoff. Please note that for the section, we evaluate the model accuracy with `Math-verify`[1], which achieves more robust

---

[1] https://github.com/huggingface/Math-Verify

Table 6: Performance comparison between THINKPRUNE and length-based rewards.

| | Accuracy | | | | | Generation Length | | | | |
|---|---|---|---|---|---|---|---|---|---|---|
| | MATH 500 | AIME | AMC | Olympiad Bench | Avg. | MATH 500 | AIME | AMC | Olympiad Bench | Avg. |
| DeepSeek-R1-Distill-Qwen-1.5B | | | | | | | | | | |
| Original Model | 85.1 | 28.1 | 73.2 | 52.1 | 59.6 | 5542 | 16553 | 9507 | 11535 | 10784 |
| EFFICIENTREASON $\alpha = 0.1$ | 84.7 | 32.1 | 72.7 | 52.4 | 60.5 | 3178 | 11135 | 5651 | 7719 | 6920 |
| THINKPRUNE-4k | 83.8 | 29.0 | 73.6 | 46.5 | 58.2 | 2709 | 8301 | 4388 | 5529 | 5232 |
| Kimi-1.5 | 83.0 | 27.1 | 72.5 | 44.3 | 56.7 | 2108 | 6617 | 3519 | 4354 | 4150 |
| EFFICIENTREASON $\alpha = 0.1$ | 82.9 | 27.8 | 70.5 | 50.3 | 57.9 | 2281 | 9525 | 4218 | 5689 | 5428 |
| THINKPRUNE-4k$\rightarrow$ 3k$\rightarrow$ 2k | 83.2 | 27.1 | 73.2 | 46.2 | 57.4 | 1938 | 5631 | 3039 | 3687 | 3574 |
| DeepSeek-R1-Distill-Qwen-7B | | | | | | | | | | |
| Original Model | 93.6 | 54.9 | 89.6 | 67.8 | 76.5 | 4008 | 13222 | 6623 | 8871 | 8181 |
| Kimi-1.5 | 92.8 | 52.9 | 88.9 | 65.8 | 75.1 | 2637 | 9633 | 4204 | 6127 | 5650 |
| EFFICIENTREASON $\alpha = 0.1$ | 93.1 | 52.3 | 88.4 | 66.4 | 75.1 | 2839 | 10660 | 4706 | 6890 | 6274 |
| EFFICIENTREASON $\alpha = 0.4$ | 93.1 | 54.1 | 89.3 | 66.8 | 75.8 | 2558 | 10106 | 4260 | 6482 | 5852 |
| THINKPRUNE-4k | 93.3 | 52.9 | 89.8 | 67.0 | 75.8 | 2929 | 9824 | 4504 | 6229 | 5872 |
| THINKPRUNE-4k$\rightarrow$ 3k$\rightarrow$ 2k | 93.3 | 51.7 | 89.1 | 66.1 | 75.1 | 2464 | 8008 | 3734 | 5123 | 4832 |

A regular hexagon can be divided into six equilateral triangles.
If the perimeter of one of the triangles is 21 inches, what is the perimeter, in inches, of the regular hexagon?

**Original Model**

*Problem understanding: 64 tokens*
Okay, so I have this problem here: … Let me try to figure this out step by step.

*Problem-solving attempt-1: 441 tokens*
First, let me visualize what's going on …

*Self-verification-1: 100 tokens*
So, is that it? … let me double-check …

*Self-verification-2: 103 tokens*
Let me confirm with another approach …

*Self-verification-3: 196 tokens*
Wait, another thought: …

*Self-verification-4: 112 tokens*
Alternatively, if I think about the height of the triangle

**ThinkPrune**

*Problem understanding: 61 tokens*
Okay, so I have this problem here: … let me try to figure this out step by step …

*Problem-solving attempt-1: 230 tokens*
First, I know a regular hexgon…
Given that the perimeter….
Now a regular hexagon has six sides. So to find …
Since each side …
Perimeter of hexagon = … = 42 inches.

*Self-verification: 57 tokens*
Wait, that seems straightforward. Let me just double-check …
Yep, that seems correct.

Figure 6: Example reasoning trace of Distill-R1-1.5B before and after length pruning on a Math-500 question.

judgment compared to other evaluators. We did not update the numbers in the main paper to maintain consistency with the previous versions of this paper.

We highlight that our method provides better accuracy–length trade-off. Compared to both two baselines, our method achieves either higher or comparable accuracy. At the same time, the average response length of our method is 30% shorter than the best baseline on DeepSeek-R1-Distill-Qwen-1.5B and 15% shorter on DeepSeek-R1-Distill-Qwen-7B.

## A.4 Additional Examples

In this section, we include additional reasoning trace examples of Distill-R1-1.5B LLM before and after applying THINKPRUNE on Math-500 questions. As shown in Figure 6 and Figure 7, the original model repeatedly checks its previous reasoning for these simple math questions, leading to many redundant self-reflection steps. On the contrary, our method successfully removes these repeated steps, and helps the LLM focus on the problem solving, while keeping perfect readability.

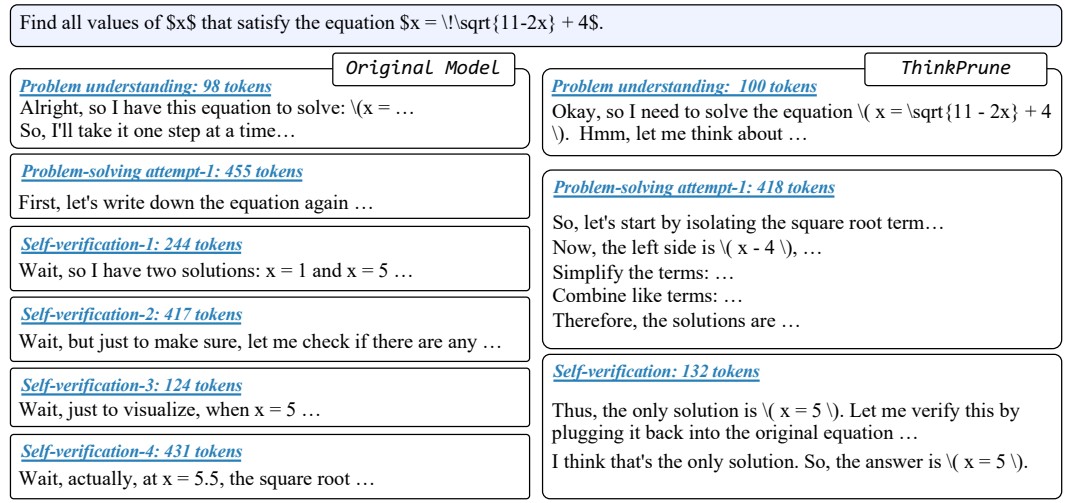

Figure 7: Example reasoning trace of Distill-R1-1.5B before and after length pruning on a Math-500 question.

Table 7: Prompt for the analysis of the reasoning behavior.

---

### **Task Description**

Given a mathematical question and its detailed solution, the task is to segment the solution into high-level problem-solving phases. The goal is to group consecutive steps into meaningful phases and output only the start and end steps of each phase.

Note: Each **reasoning step** in the solution is separated by a **double line break ("\n\n")**.
— — — — — —
### **Requirements**

1. **Segment the full solution into distinct problem-solving phases** based on logical progression.
2. **Each phase should have a start and an end step**.
3. **A phase can appear multiple times** in different parts of the solution.
4. **The order of phases is flexible**—they can appear in any logical sequence depending on the nature of the solution.
5. **Only the first and last steps of each phase should be output**, reducing redundancy.

— — — — — —
### **High-Level Problem-Solving Phases**

Each step in the solution should belong to one of the following **ten high-level phases**:

1. **Understanding the Problem**: Identifying given data, definitions, and the goal.
2. **Reformulating the Problem**: Changing variables, rewriting expressions, or restructuring sums.
3. **Applying Known Theorems/Properties**: Using standard formulas, identities, or mathematical principles.
4. **Breaking Down into Subproblems**: Decomposing the problem into manageable components.
5. **Computing or Simplifying Expressions**: Performing algebraic manipulation or numerical evaluation.
6. **Substituting Known Values or Results**: Using precomputed values or standard mathematical constants.
7. **Reassess and Verify Local Steps**: Checking for errors or inconsistencies within a small part of the solution.
8. **Reassess the Whole Solution**: Reviewing the entire solution for logical correctness and consistency.
9. **Exploring Alternative Approaches**: Considering different methods to solve the problem.
10. **Finalize and Present the Answer**: Writing the final result and ensuring clarity.
— — — — — —
### **Output Format**
The output should consist of **multiple phases**, each represented in the following format:

```
[ Phase X ]: {Phase Name}
[Start]: {Text of first step in the phase}
[End]: {Text of last step in the phase}
```

Where:

- **Phase X** represents the index of the phase (e.g., Phase 1, Phase 2, etc.).
- **Phase Name** is one of the ten high-level categories.
- **Start** is the first step of the phase.
- **End** is the last step of the phase.

#### **Example Output**

```
[Phase 1]: Understanding the Problem
[Start]: [Text of step 1]
[End]: [Text of step 3]

[Phase 2]: Reformulating the Problem
[Start]: [Text of step 4]
[End]: [Text of step 5]
...
[Phase 4]: Finalize and Present the Answer
[Start]: [Text of step X]
[End]: [Text of step Y]
```

