# OpenReview forum: "ThinkPrune: Pruning Long Chain-of-Thought of LLMs via Reinforcement Learning"
_TMLR — Accepted by TMLR_

### Review · Reviewer_CHmk · 2025-08-27

**Summary Of Contributions:**

ThinkPrune adds a hard length cap during RL (GRPO) and iteratively tightens it (e.g., 4k→3k→2k) so the model learns to prune redundant reasoning while preserving accuracy; shows shorter chains at similar performance on math benchmarks.

**Strengths**

The paper is well written and easy to follow. The proposed idea is simple yet effective, and can be directly applied to existing reasoning models to improve efficiency.

**Weakness**

**[W1] Insufficient baselines**

Over-thinking mitigation is already well studied, particularly in R1-distilled reasoning models. To make the contribution convincing, the paper should compare against stronger or closer baselines such as O1-Pruner [1] and efficient length-aware training methods [2]. Without these, it is difficult to disentangle improvements from the method itself versus the weakness of the selected baselines.

[1] O1-Pruner: Length-Harmonizing Fine-Tuning for O1-Like Reasoning Pruning, 2025

[2] Training Large Language Models to Reason Efficiently, 2025

**[W2] Setup concerns**

Training uses AIME/AMC subsets, while evaluation includes AIME24/AMC23, which are likely from a similar distribution. Please provide cross-dataset evaluation (e.g., train on AIME/AMC) and consider a different reasoning domain (e.g., GPQA) to confirm generalization.

**[W3] Implementation details to clarify**

The paper lacks key implementation details (e.g., training steps per RL rounds)
Are these hyper-parameters tuned separately per model (e.g., for Qwen-1.5B vs other backbones)?
If so, please specify the tuning ranges and selection criteria to ensure reproducibility.

**Audience:**

Yes

**Audience Explanation:**

Potentially yes, but it is somewhat unclear. The paper needs to better justify its advantages over existing methods in order to clearly establish its contribution and relevance for the audience.

**Claims And Evidence:**

Yes

**Claims Explanation:**

Overall, the claims are well supported through a diverse set of experiments and models. However, the setup-related weakness (i.e., [W2]) requires further clarification to fully validate the conclusions.

**Requested Changes:**

See Weakness.

---

> ### Author Response · Authors · 2025-10-19
>
> We thank Reviewer CHmk for the support and the valuable suggestions for our paper. Please see our response to the questions and major comments below.
>
> **Q1: Cross-dataset evaluation to further demonstrate the generalization of the proposed method.**
>
> Thank you very much for this insightful suggestion. To further verify the generalization ability of out method beyond math reasoning tasks, we additionally evaluate our method on **GPQA**, **MMLU**, and **MMLU-Pro**, which cover broader domains such as science and general knowledge. Importantly, all models here are **trained only on math question**. The new results are summarized below. We also include the average accuracy and length on math benchmarks for your convenience.
>
> `DeepSeek-R1-Distill-Qwen-1.5B`
>
> | | Accuracy |||   | Length | ||   |
> | ----------------- | -------- | ---- | ---- | ------- | ------ | ----- | ---- | ------- |
> | | Math     | GPQA | MMLU | MMLUPro | Math   | GPQA  | MMLU | MMLUPro |
> | Original    | 56.8     | 32.7 | 49.1 | 39.4    | 10355  | 10072 | 2816 | 5661    |
> | ThinkPrune-4k     | 59.9     | 34.5 | 48.8 | 39.9    | 5247   | 6465  | 1685 | 3515    |
> | ThinkPrune-4k$\rightarrow$3k  | 59.3     | 35.6 | 48.8 | 39.4    | 4433   | 5349  | 1395 | 2982    |
> | ThinkPrune-4k$\rightarrow$3k $\rightarrow$ 2k | 59.4     | 34.2 | 48.3 | 39.7    | 3668   | 4639  | 1243 | 2687    |
>
>
>
> `DeepSeek-R1-Distill-Qwen-7B`
>
> | | Accuracy |||   | Length |||   |
> | ----------------- | -------- | ---- | ---- | ------- | ------ | ---- | ---- | ------- |
> | | Math     | GPQA | MMLU | MMLUPro | Math   | GPQA | MMLU | MMLUPro |
> | Original    | 76.5     | 49.8 | 67.8 | 58.6    | 8681   | 8330 | 2237 | 4254    |
> | ThinkPrune-4k     | 75.8     | 50.3 | 67.3 | 59.1    | 5872   | 6815 | 1828 | 3456    |
> | ThinkPrune-4k$\rightarrow$3k  | 75.5     | 50.1 | 67.2 | 59.5    | 5277   | 6317 | 1756 | 3269    |
> | ThinkPrune-4k$\rightarrow$3k $\rightarrow$ 2k | 75.1     | 49.4 | 67.1 | 59.4    | 4832   | 6082 | 1615 | 3079    |
>
> We highlight the following key observations:
>
> - ThinkPrune can natually generalize to non-math domains despite being **trained exclusively on math reasoning data**. On all benchmarks, there is only minimal differences (within **±1.5%**) in accuracy compared to the original models.
> - The **reasoning length is significantly reduced across all benchmarks**. For the 1.5B model, *ThinkPrune-iter2k* shortens the reasoning chain by **55–65%** (e.g., from 10,355 → 3,574 tokens on Math, and from 10,072 → 4,639 on GPQA), while maintaining nearly identical accuracy. The 7B model shows similar trends, with **35–45% shorter reasoning** and stable or slightly improved performance.
>
> These results further confirm the generailization ability of our method, which can reduces redundancy while maintains the performance even on unseen domains. We will include these new results and analyses in the final version.

---

> ### Author Response · Authors · 2025-10-19
>
> **Q2: Performance comparison with other length-reduction methods.**
>
> We thank the reviewer for highlighting the importance of stronger baselines. Following your suggestion, we conducted additional experiments with two representative RL-based length-reduction methods:
>
> - EfficientReason [1], which applies a group-relative length penalty with a tunable coefficient $\alpha$ controlling the strength of the penalty.
> - Kimi-1.5 [2], which also penalizes longer responses based on their relative lengths.
>
> We further extended the evaluation to a larger model, **DeepSeek-R1-Distill-Qwen-7B**, to provide a more comprehensive assessment. For our method, we report two configurations
>
> - ThinkPrune-4k: models are directly trained with a 4k token length limit
> - ThinkPrune-4k$\rightarrow$3k $\rightarrow 2k$: iterative pruning by decreasing the length limit from 4k to 3k and 2k.
>
>  The experiment results are as below. To better demonstrate the trade-off between reasoning length and performance, we group models with similar accuracies.
>
> `deepseek-ai/DeepSeek-R1-Distill-Qwen-1.5B`:
>
> |     | Accuracy |||   |    | Length  | ||   |    |
> | --------------------------------------------- | -------- | ---- | ---- | ------------- | -------- | ------- | ----- | ---- | ------------- | -------- |
> |     | Math500  | AIME | AMC  | OlympiadBench | **Avg.** | Math500 | AIME  | AMC  | OlympiadBench | **Avg.** |
> | Original Model  | 85.1     | 28.1 | 73.2 | 52.1    | 59.6     | 5542    | 16553 | 9507 | 11535   | 10784    |
> |     |    |||   |    |   | ||   |    |
> | EfficientReason $\alpha=0.1$| 84.7     | 32.1 | 72.7 | 52.4    | 60.5     | 3178    | 11135 | 5651 | 7719    | 6920     |
> | ThinkPrune-4k   | 84.8     | 29.4 | 73.1 | 52.3    | 59.9     | 2752    | 8421  | 4278 | 5539    | 5247     |
> |     |    |||   |    |   | ||   |    |
> | Kimi-1.5  | 83.0     | 27.1 | 72.5 | 44.3    | 56.7     | 2108    | 6617  | 3519 | 4354    | 4150     |
> | EfficientReason $\alpha=0.4$| 82.9     | 27.8 | 70.5 | 50.3    | 59.3     | 2281    | 9525  | 4218 | 5689    | 5428     |
> | ThinkPrune-4k$\rightarrow$3k $\rightarrow 2k$ | 84.3     | 28.3 | 72.7 | 52.2    | 59.4     | 1966    | 6004  | 3020 | 3683    | 3668     |
>
>
>
> `deepseek-ai/DeepSeek-R1-Distill-Qwen-7B`:
>
> |     | Accuracy |||   |    | Length  | ||   |    |
> | --------------------------------------------- | -------- | ---- | ---- | ------------- | -------- | ------- | ----- | ---- | ------------- | -------- |
> |     | Math500  | AIME | AMC  | OlympiadBench | **Avg.** | Math500 | AIME  | AMC  | OlympiadBench | **Avg.** |
> | Original Model  | 93.6     | 54.9 | 89.6 | 67.8    | 76.5     | 4008    | 13222 | 6623 | 8871    | 8181     |
> |     |    |||   |    |   | ||   |    |
> | Kimi-1.5  | 92.8     | 52.9 | 88.9 | 65.8    | 75.1     | 2637    | 9633  | 4204 | 6127    | 5650     |
> | EfficientReason $\alpha=0.1$| 93.1     | 52.3 | 88.4 | 66.4    | 75.1     | 2839    | 10660 | 4706 | 6890    | 6274     |
> | EfficientReason $\alpha=0.4$| 93.1     | 54.1 | 89.3 | 66.8    | 75.8     | 2558    | 10106 | 4260 | 6482    | 5852     |
> | ThinkPrune-4k   | 93.3     | 52.9 | 89.8 | 67.0    | 75.8     | 2929    | 9824  | 4504 | 6229    | 5872     |
> | ThinkPrune-4k$\rightarrow$3k $\rightarrow 2k$ | 93.3     | 51.7 | 89.1 | 66.1    | 75.1     | 2464    | 8008  | 3734 | 5123    | 4832     |
>
> We highlight that our method consistently achieves a **better accuracy–length trade-off** than both baselines. At comparable accuracy levels, the reasoning length of ThinkPrune is about **30% shorter** than the best baseline on the 1.5B model and **15% shorter** on the 7B model. These results confirm that our method provides a simple yet more effective way to reduce reasoning length compared to existing length-penalty-based methods. We will include these new results and analysis in the final revision.
>
>
>
> [1] Arora, Daman, and Andrea Zanette. "Training language models to reason efficiently." arXiv preprint arXiv:2502.04463 (2025).
>
> [2] Team, Kimi, et al. "Kimi k1. 5: Scaling reinforcement learning with llms." arXiv preprint arXiv:2501.12599 (2025).=

---

> > ### Comment · Reviewer_CHmk · 2025-10-28
> >
> > Thank you for addressing my earlier concerns regarding generalizability and the inclusion of stronger baselines. However, the issue of reproducibility remains unresolved. In particular, it is still unclear how training steps are defined and executed per RL round.

---

> > > ### Author Response · Authors · 2025-10-28
> > >
> > > **Question: How training steps are defined and executed per RL round?**
> > >
> > > Thank you very much for this important question. We realized that in our previous response, we focused on providing detailed explanations and results for Q1 and Q2, and inadvertently did not include the answer to this question. We sincerely apologize for the oversight. We are glad that this point was also raised by other reviewers (https://openreview.net/forum?id=V51gPu1uQD&noteId=07fgGUQZcT), and we provide the complete clarification below, consistent across all responses.
> > >
> > > - **Stop criterion**.  We would like to first explain the consistent stopping criterion we adopted for the RL round. Since our main goal is to reduce the reasoning length of LLMs **while preserving their performance**, we monitor validation accuracy on AIME 2022 and AIME 2023 to determine when to stop training. Specifically, we allow up to a **10 % relative drop in pass@1 accuracy** on AIME 22 + 23 compared with the original model in each RL round. A given training round (e.g., RL 4k or 4k $\rightarrow$ 3k) stops when the performance drop exceeds 10% or the model reaches the maximum training steps (**10 epochs** in our setting). These details are described in Section 3.3 of the main paper, and we will highlight them more clearly in the final revision.
> > >
> > > - **Training steps per iteration.** The actual steps used for each RL round are summarized below.
> > >
> > >   - `DeepSeek-R1-Distill-Qwen-1.5B` and `DeepScaleR-1.5B-Preview`: We observe the performance degradation in each iteration is mild, therefore, we train for **10 epochs (180 steps)** at all RL rounds, including RL 4k, $4k \rightarrow 3k$, and  $3k\rightarrow$2k . For example, :the validation performance (AIME 22 + 23) during RL 4k training on `DeepSeek-R1-Distill-Qwen-1.5B` is shown below:
> > >
> > >     | Step  | Validation Pass@1 | Average Length |
> > >     | ----------------------- | ----------------- | -------------- |
> > >     | 20    | 20.7  | 17712    |
> > >     | 40    | 21.1  | 16519    |
> > >     | 60    | 19.8  | 14825    |
> > >     | 80    | 20.8  | 13251    |
> > >     | 100   | 20.8  | 11849    |
> > >     | 120   | 21.0  | 11063    |
> > >     | 140   | 20.9  | 10251    |
> > >     | 160   | 20.3  | 9919     |
> > >     | 180 (epoch 10 finished) | 20.6  | 9336     |
> > >
> > >     The performance drop is minimal, so we use the last checkpoint to start the next stage (4k → 3k). Similarly, the validation performance of the model within the training stage is:
> > >
> > >     | Step  | ValAcc | ValLen |
> > >     | ----------------------- | ------ | ------ |
> > >     | 20    | 21.7   | 9048   |
> > >     | 40    | 21.6   | 8478   |
> > >     | 60    | 22.1   | 8321   |
> > >     | 80    | 22.1   | 8280   |
> > >     | 100   | 20.3   | 7706   |
> > >     | 120   | 19.6   | 7849   |
> > >     | 140   | 20.8   | 7636   |
> > >     | 160   | 20.3   | 7760   |
> > >     | 180 (epoch 10 finished) | 20.5   | 7402   |
> > >
> > >     Therefore, we proceed to the $3k \rightarrow 2k$ experiment from the last checkpoint.
> > >
> > >   - `QwQ-32B`: For the larger model, we observe faster performance changes. During the first RL round with 4k token limit (RL 4k), the validation accuracy on AIME 22 + 23 decreases quickly:
> > >
> > >   | Step | ValAcc | ValLen |
> > >   | ---- | ------ | ------ |
> > >   | 20   | 76.4   | 12707  |
> > >   | 40   | 74.7   | 9841   |
> > >   | 60   | 70.9   | 8524   |
> > >
> > > - Therefore, we stop at **step 40** and use this checkpoint to launch the next round (4k → 3k), which in turn is stopped at **step 20** for similar reasons.
> > >
> > > We will include these details results in our final revision. Thank you very much for your suggestion.

---

> > > > ### Comment · Reviewer_CHmk · 2025-10-28
> > > >
> > > > Thank you for the clarifications. My concerns have been adequately addressed.

---

### Review · Reviewer_PkdD · 2025-09-30

**Summary Of Contributions:**

This paper presents ThinkPrune, and RL based method to prune the CoT for LLM reasoning. Experimental results suggest that the proposed method is able to significantly reduce the thinking length of different LLMs on common math reasoning benchmarks without drastic sacrifice of the model accuracy.

**Audience:**

Yes

**Audience Explanation:**

Reducing thinking length is an interesting research topic for LLM reasoning and will be interested to TMLR's audience.

**Claims And Evidence:**

No

**Claims Explanation:**

- Details of 4k->3k and 4k->3k->2k is missing. Do you use the same amount of iterations for the two experiments or same amount of iterations for each length (e.g. 1000 iterations for 4k and another 1000 for 3k (a total of 2000 iterations) vs 1000 iterations for 4/3/2k (a total of 3000 iterations)).
- Figure 2, RL4K and RL3K definition is missing.
- "This suggests that redundancy in reasoning is related to problem difficulty and that length pruning can adaptively remove unnecessary thinking for questions at different difficulties." -> How do the authors defined unnecessary thinking?
- Missing explanation of the performance drop at average thinking token 1000 in Figure 3 DeepScaleR AIME

**Requested Changes:**

- Presentation of this paper needs to be significantly improved.
  - Please add the GRPO objective for readers.
  - Table 1 is very crowded. To me there are three comparisons in Table 1: 1. original & one shot (row 1/2/3/4), 2. original & iterative (row 1/5/6), 3. one shot vs iterative at their corresponding lengths (3 vs 5, 4 vs 6). I would suggest color code the Table with two different colors (one shot and iterative) and color code the text. So that it is clearer for the reader which comparison you are focusing on at each paragraph.
  - Please highlight the takeaways from the experiments more clearly.
- It is unclear to me the goal of the paper. I think it's a cool technique to reduce the thinking lengths while maintaining the accuracy but the desired outcome of such reduced budget is not discussed by the authors. For example, it's unclear whether reducing the length provides us more budget that can correctly answer questions that we cannot previously? It's unclear whether the ability to reduce length relates to the difficulty of the problem? I think these are critical parts that needs to be shown experimentally or at least discussed in the paper.
- Nit: p5 GROP -> GRPO

---

> ### Author Response · Authors · 2025-10-19
>
> We thank Reviewer PkdD for the support and the valuable suggestions for our paper. Please see our response to the questions and major comments below.
>
> **Q1: Details of training steps used in 4k->3k and 4k->3k->2k experiments.**
>
> We thank the reviewer for raising this point. We agree that the stopping criterion for each stage of iterative length pruning is important. In what follows, we clarify both our **stopping criteria** and the **actual training steps** below.
>
> - **Stop criterion**.  Since our main goal is to reduce the reasoning length of LLMs **while preserving their performance**, we monitor validation accuracy on AIME 2022 and AIME 2023 to determine when to stop training. Specifically, we allow up to a **10 % relative drop in pass@1 accuracy** on AIME 22 + 23 compared with the original model in each iteration. A given training stage (e.g., RL 4k or 4k $\rightarrow$ 3k) stops when the performance drop exceeds 10% or the model reaches the maximum training steps (**10 epochs** in our setting). sThese details are described in Section 3.3 of the main paper, and we will highlight them more clearly in the final revision.
>
> - **Training steps per iteration.** The actual steps used for each model are summarized below.
>
>   - `DeepSeek-R1-Distill-Qwen-1.5B` and `DeepScaleR-1.5B-Preview`: We observe the performance degradation in each iteration is mild, therefore, we train for **10 epochs (180 steps)** at all stages, including RL 4k, $4k \rightarrow 3k$, and  $3k\rightarrow$2k . For example, :the validation performance (AIME 22 + 23) during RL 4k training on `DeepSeek-R1-Distill-Qwen-1.5B` is shown below:
>
>     | Step  | Validation Pass@1 | Average Length |
>     | ----------------------- | ----------------- | -------------- |
>     | 20    | 20.7  | 17712    |
>     | 40    | 21.1  | 16519    |
>     | 60    | 19.8  | 14825    |
>     | 80    | 20.8  | 13251    |
>     | 100   | 20.8  | 11849    |
>     | 120   | 21.0  | 11063    |
>     | 140   | 20.9  | 10251    |
>     | 160   | 20.3  | 9919     |
>     | 180 (epoch 10 finished) | 20.6  | 9336     |
>
>     The performance drop is minimal, so we use the last checkpoint to start the next stage (4k → 3k). Similarly, the validation performance of the model within the training stage is:
>
>     | Step  | ValAcc | ValLen |
>     | ----------------------- | ------ | ------ |
>     | 20    | 21.7   | 9048   |
>     | 40    | 21.6   | 8478   |
>     | 60    | 22.1   | 8321   |
>     | 80    | 22.1   | 8280   |
>     | 100   | 20.3   | 7706   |
>     | 120   | 19.6   | 7849   |
>     | 140   | 20.8   | 7636   |
>     | 160   | 20.3   | 7760   |
>     | 180 (epoch 10 finished) | 20.5   | 7402   |
>
>     Therefore, we proceed to the $3k \rightarrow 2k$ experiment from the last checkpoint.
>
>   - `QwQ-32B`: For the larger model, we observe faster performance changes. During the first stage (RL 4k), the validation accuracy on AIME 22 + 23 decreases quickly:
>
>   | Step | ValAcc | ValLen |
>   | ---- | ------ | ------ |
>   | 20   | 76.4   | 12707  |
>   | 40   | 74.7   | 9841   |
>   | 60   | 70.9   | 8524   |
>
> - Therefore, we stop at **step 40** and use this checkpoint to launch the next stage (4k → 3k), which in turn is stopped at **step 20** for similar reasons.
>
> We will include these details results in our final revision. Thank you very much for your suggestion.

---

> ### Author Response · Authors · 2025-10-19
>
> **Q2: Why does the performance drop at average thinking token 1000 in Figure 3 DeepScaleR AIME?**
>
> We appreciate the reviewer’s careful observation. This phenomenon is empirically observed, and we provide the following clarification based on our analysis of the model outputs.
>
> - For existing thinking LLMs such as **DeepSeek-R1** [1] and **GPT-oss** [2], each model output consists of two segments: a **reasoning (thinking)** segment and an **answer** segment. Following the original *budget-forcing* implementation [3], we **only apply the token budget constraint to the reasoning segment**, while the answer segment remains unconstrained.
> - When the reasoning budget is **very small (e.g., 500 tokens)**, the model often continues part of its reasoning within the **answer segment** after the thinking phase is cut off. Because the answer portion is not length-limited, this behavior effectively allows the model to “cheat” beyond the reasoning budget, occasionally resulting in unexpectedly high accuracy despite the short budget.
> - At a **moderate budget (around 1000 tokens)**, such “cheating” occurs less frequently, which leads to a temporary drop in performance.
>    For larger budgets (e.g., 2000–4000 tokens), the model again has sufficient reasoning space, and performance recovers accordingly.
>
> We also note that this phenomenon appears only for the **DeepScaleR** model, and may be coincidental.
>  In short, the dip at 1000 tokens likely results from **interaction effects between reasoning truncation and the unconstrained answer segment**, rather than any issue with our evaluation procedure. We will clarify this observation and its possible cause in the revised paper.
>
> [1] Guo, Daya, et al. "Deepseek-r1: Incentivizing reasoning capability in llms via reinforcement learning." arXiv preprint arXiv:2501.12948 (2025).
>
> [2] Agarwal, Sandhini, et al. "gpt-oss-120b & gpt-oss-20b model card." *arXiv preprint arXiv:2508.10925* (2025).
>
> [3] Muennighoff, Niklas, et al. "s1: Simple test-time scaling." arXiv preprint arXiv:2501.19393 (2025).
>
>
>
> **Q3: How do the authors defined unnecessary thinking?**
>
> Thank you for raising this question. We did not explicitly define *unnecessary thinking* in a formal sense, and we acknowledge that this term may have been ambiguous in the original draft. Here we clarify our intended meaning.
>
> - In our experiments, we observe that on **relatively easy benchmarks** such as **MATH500**, our method can reduce the reasoning length by up to 60% while maintaining almost identical accuracy compared with the original model. This suggests that a substantial portion of the generated reasoning steps does not contribute to the correctness of the final answer.
> - Therefore, we refer to unnecessary thinking as redundant reasoning content that can be removed **without changing the correctness of the model’s output**. This notion is **empirical rather than formal**.it simply reflects the observation that large reasoning models often produce verbose and repetitive intermediate explanations.
>
> We will clarify this interpretation in the revised paper to avoid confusion.

---

> ### Author Response · Authors · 2025-10-19
>
> **Q4: What is the goal of the paper?**
>
> Thank you for the helpful question. The goal of our paper is **two-fold**:
>
> - First, **we aime to propose an effective length reduction method compared with existing approaches** such as budget-forcing [1] and length-based penalty methods [2,3].
>   - Compared to budget-forcing: Budget-forcing truncates outputs only at inference time, whereas ThinkPrune is applied during training, enabling the model to actively learn more efficient reasoning patterns instead of passively adapting to truncated outputs.
>   - Compared to length-based penalties: Length-penalty methods require manual design of penalty terms and careful tuning of their strengths. In contrast, our method introduces **no additional hyperparameters** and allows **direct, explicit control** of generation length via a simple maximum token limit. This design is both intuitive and easy to adjust for deployment, while achieving better accuracy-length trade-offs than prior RL-based methods such as Kimi-1.5 [2]. We additionally include another length-penalty baseline,  EfficientReason [3] for comparison. The results with backbone model `deepseek-ai/DeepSeek-R1-Distill-Qwen-1.5B` are as below:
>
> |     | Accuracy |||   |    | Length  | ||   |    |
> | --------------------------------------------- | -------- | ---- | ---- | ------------- | -------- | ------- | ----- | ---- | ------------- | -------- |
> |     | Math500  | AIME | AMC  | OlympiadBench | **Avg.** | Math500 | AIME  | AMC  | OlympiadBench | **Avg.** |
> | Original Model  | 85.1     | 28.1 | 73.2 | 52.1    | 59.6     | 5542    | 16553 | 9507 | 11535   | 10784    |
> | Kimi-1.5  | 83.0     | 27.1 | 72.5 | 44.3    | 56.7     | 2108    | 6617  | 3519 | 4354    | 4150     |
> | EfficientReason | 82.9     | 27.8 | 70.5 | 50.3    | 57.9     | 2281    | 9525  | 4218 | 5689    | 5428     |
> | ThinkPrune-4k$\rightarrow$3k $\rightarrow 2k$ | 84.3     | 28.3 | 72.7 | 52.2    | 59.4     | 1966    | 6004  | 3020 | 3683    | 3668     |
>
> where we see our method provides shorter responses while maintains similar performance compare to baseline methods.
>
> - Second, **we aim to reveal how much the reasoning length of LLMs can be pruned without degrading performance.** This second goal aims to provide empirical insights into the **capability boundaries of current reasoning LLMs**. For instance, we find that for smaller-scale models, we can maintain nearly the same accuracy as the original model while reducing reasoning length by **50-60%**, whereas for the larger **QwQ-32B** model, the length can be reduced by about **30 %** before noticeable performance degradation occurs.
>
> We will make these two objectives clearer in the revised manuscript.
>
> [1] Muennighoff, Niklas, et al. "s1: Simple test-time scaling." arXiv preprint arXiv:2501.19393 (2025).
>
> [2] Team, Kimi, et al. "Kimi k1. 5: Scaling reinforcement learning with llms." arXiv preprint arXiv:2501.12599 (2025).
>
> [3] Arora, Daman, and Andrea Zanette. "Training language models to reason efficiently." arXiv preprint arXiv:2502.04463 (2025).
>
>
>
>
> **Q5: Presentation issues with the paper.**
>
> Thank you very much for the constructive suggestions, which will help us further improve the clarity and readability of our paper. We will include the GRPO training objective for clarity and revise Table 1 to make the comparisons between the one-shot and iterative settings easier to follow. At the meantime, We would also like to highlight the key takeaways of our experiments as follows:
>
> - Applying our method for length reduction effectively reduces the reasoning length of existing thinking LLMs by **40–60 %** with minimal performance degradation.
> - Iteratively lowering the length constraint further shortens the reasoning length while maintaining strong accuracy.
> - Compared with length-penalty-based baselines, ThinkPrune achieves the same or better performance with shorter reasoning length than EfficientReason and Kimi-1.5.
>
> We will include your comments in the final revision and add the takeways into our paper.

---

> > ### Comment · Reviewer_PkdD · 2025-11-19
> >
> > Thank you for your detailed response. Most of my questions have been resolved.

---

### Review · Reviewer_aPeP · 2025-10-05

**Summary Of Contributions:**

In this work, the authors propose ThinkPrune, a reinforcement learning (RL) framework designed to regularize the output length of long chain-of-thought (CoT) reasoning. Specifically, they introduce a strict budget constraint in the reward function, where reasoning outputs that exceed the budget receive a reward of zero. Through extensive experiments, the authors demonstrate that ThinkPrune achieves a better budget–accuracy trade-off than existing budget-forcing methods on the original model. They also present a detailed analysis of the resulting reasoning behavior.

**Audience:**

Yes

**Audience Explanation:**

Reducing the length of chain-of-thought reasoning is an important and actively studied problem. The authors present a simple yet effective method to address this issue, and I believe the findings will be of interest to at least some readers in the community.

**Broader Impact Concerns:**

I have no concerns about the ethical implications of the work.

**Claims And Evidence:**

Yes

**Claims Explanation:**

While I am leaning towards the answer 'yes' to the above questions, I have some remaining concerns. I will present my view by suggesting the strengths and weaknesses of the paper:

**Strengths**
1. The paper is clearly written and easy to follow.
2. The approach is evaluated across three different models, supporting the generalizability and robustness of the contribution.
3. The authors conduct thorough analyses of reasoning behavior, including keyword frequency, behavioral changes, and perplexity analysis.

**Weaknesses**
1. The main concern lies in the choice of the primary baseline. As noted by the authors, an existing RL-based length regularization method (the length-based reward in Kimi-1.5) already addresses a similar goal. Thus, the significance of ThinkPrune primarily stems from its simpler reward formulation. While Table 2 includes some comparison, it is limited to the 1.5B model. A stronger justification is needed for why the baseline comparison excludes the existing RL-based method and instead focuses on the original model, throughout the paper.
2. In Section 4.2, the authors mention that ThinkPrune still produces long CoTs for difficult problems, but do not discuss how this affects performance. Could the reward design potentially harm performance on harder samples that require longer reasoning chains, since such cases may receive limited learning signals?

3. The framing in Section 2.2 is somewhat unclear. The authors state that KimiTeam et al. (2025), Chen et al. (2024), and Shen et al. (2025) propose methods used during the RL training stage, while ThinkPrune is positioned as a method for ‘further’ RL training. It would be helpful to clarify what distinguishes ThinkPrune from these prior RL-stage methods, or why it is particularly suited for a secondary RL phase.

**Requested Changes:**

I have listed all suggested changes in the Weakness section. Weakness 1 is critical to securing my recommendation, while others would strengthen the work in my view.

---

> ### Author Response · Authors · 2025-10-19
>
> We thank Reviewer aPeP for the support and the valuable suggestions for our paper. Please see our response to the questions and major comments below.
>
> **Q1: More comparison with existing RL-based approaches.**
>
> Thank you for this insightful suggestion. We agree that stronger comparisons with existing RL-based length-regularization methods can better demonstrate the effectiveness of our method. In response, we have conducted additional experiments including two representative RL-based approaches:
>
> - EfficientReason [1], which adopts a group-relative length penalty.  Their method further introduce an coefficient $\alpha$  to control the strength of the penalty.
> - Kimi-1.5 [2], which also penalize longer responses based on the relative lengths.
>
> We further extended our evaluation to a larger backbone model, **DeepSeek-R1-Distill-Qwen-7B**, to provide a more comprehensive evaluation. For our method, we report the performance with two versions:
>
> - ThinkPrune-4k: LLMs are  directly trained with a length limit of 4k output tokens.
> - ThinkPrune-4k$\rightarrow$3k $\rightarrow 2k$: Iterative length pruning by decreasing the length limit from 4k to 3k and 2k.
>
>  The experiment results are as below. We group models with similar accuracy to better demonstrate the accuracy-length tradeoff:
>
> `deepseek-ai/DeepSeek-R1-Distill-Qwen-1.5B`:
>
> || Accuracy |||   |    | Length  | ||   ||
> | --------------------------------------------- | -------- | ---- | ---- | ------------- | -------- | ------- | ----- | ---- | ------------- | -------- |
> |     | Math500  | AIME | AMC  | OlympiadBench | **Avg.** | Math500 | AIME  | AMC  | OlympiadBench | **Avg.** |
> | Original Model  | 85.1     | 28.1 | 73.2 | 52.1    | 59.6     | 5542    | 16553 | 9507 | 11535   | 10784    |
> |     |    |||   |    |   | ||   |    |
> | EfficientReason $\alpha=0.1$| 84.7     | 32.1 | 72.7 | 52.4    | 60.5     | 3178    | 11135 | 5651 | 7719    | 6920     |
> | ThinkPrune-4k   | 84.8     | 29.4 | 73.1 | 52.3    | 59.9     | 2752    | 8421  | 4278 | 5539    | 5247     |
> |     |    |||   |    |   | ||   |    |
> | Kimi-1.5  | 83.0     | 27.1 | 72.5 | 44.3    | 56.7     | 2108    | 6617  | 3519 | 4354    | 4150     |
> | EfficientReason $\alpha=0.4$| 82.9     | 27.8 | 70.5 | 50.3    | 57.9     | 2281    | 9525  | 4218 | 5689    | 5428     |
> | ThinkPrune-4k$\rightarrow$3k $\rightarrow 2k$ | 84.3     | 28.3 | 72.7 | 52.2    | 59.4     | 1966    | 6004  | 3020 | 3683    | 3668     |
>
>
>
> `deepseek-ai/DeepSeek-R1-Distill-Qwen-7B`:
>
> |     | Accuracy |||   |    | Length  | ||   |    |
> | --------------------------------------------- | -------- | ---- | ---- | ------------- | -------- | ------- | ----- | ---- | ------------- | -------- |
> |     | Math500  | AIME | AMC  | OlympiadBench | **Avg.** | Math500 | AIME  | AMC  | OlympiadBench | **Avg.** |
> | Original Model  | 93.6     | 54.9 | 89.6 | 67.8    | 76.5     | 4008    | 13222 | 6623 | 8871    | 8181     |
> |     |    |||   |    |   | ||   |    |
> | Kimi-1.5  | 92.8     | 52.9 | 88.9 | 65.8    | 75.1     | 2637    | 9633  | 4204 | 6127    | 5650     |
> | EfficientReason $\alpha=0.1$| 93.1     | 52.3 | 88.4 | 66.4    | 75.1     | 2839    | 10660 | 4706 | 6890    | 6274     |
> | EfficientReason $\alpha=0.4$| 93.1     | 54.1 | 89.3 | 66.8    | 75.8     | 2558    | 10106 | 4260 | 6482    | 5852     |
> | ThinkPrune-4k   | 93.3     | 52.9 | 89.8 | 67.0    | 75.8     | 2929    | 9824  | 4504 | 6229    | 5872     |
> | ThinkPrune-4k$\rightarrow$3k $\rightarrow 2k$ | 93.3     | 51.7 | 89.1 | 66.1    | 75.1     | 2464    | 8008  | 3734 | 5123    | 4832     |
>
> We highlight that our method provides **better accuracy–length trade-off**. Compared to both two baselines, our method achieves either higher or comparable accuracy. At the same time, the average response length of our method is 30% shorter than the best baseline on  `deepseek-ai/DeepSeek-R1-Distill-Qwen-1.5B` and 15% shorter on `deepseek-ai/DeepSeek-R1-Distill-Qwen-7B`. We will include these new results and analysis into our final version of the paper.
>
>
>
> [1] Arora, Daman, and Andrea Zanette. "Training language models to reason efficiently." arXiv preprint arXiv:2502.04463 (2025).
>
> [2] Team, Kimi, et al. "Kimi k1. 5: Scaling reinforcement learning with llms." arXiv preprint arXiv:2501.12599 (2025).

---

> ### Author Response · Authors · 2025-10-19
>
> **Q2:  Will the reward design potentially harm performance on harder samples that require longer reasoning chains?**
>
> We thank the reviewer for raising this important point. Currently, the length constraint is applied to all questions regardless of the difficulty levels because the primary goal of our paper is to **quantify how much reasoning can be pruned before accuracy degrades**, rather than to optimize for per-sample efficiency. Nevertheless, we would like to discuss the potential effects of this design and possible extensions that could further improve our method.
>
> - **Effect of GRPO’s gradient mechanism.** Under the GRPO training framework, when all sampled responses for a given question exceed the length limit, each receives a clipped reward of zero, which results in **no gradient update** for that instance. Therefore, for those hard examples during training (where all responses are likely to exceed the length limit and receive rewards of 0), the model does not receive a misleading signal that would force it to shorten its reasoning on inherently difficult problems. As a result, these hard cases are effectively left unchanged during training rather than incorrectly penalized.
> - **Empirical behavior on challenging datasets.** Our experiments show that our method produces longer reasoning chains and maintains strong performance on difficult benchmarks(*e.g.*, AIME and OlympiadBench). This indicates that the model can still allocate longer reasoning traces when needed while pruning redundant reasoning elsewhere..
> -  **Extension of adaptive, difficulty-aware length constraint .** We have seen some related work that explores RL-based training on datasets with difficulty level annotations [1, 2]. Since automatically estimating or labeling question difficulty in a reliable and scalable way is a non-trivial challenge, exploring this direction is beyond the scope of our current study.
>
> [1] Abdin, Marah, et al. "Phi-4-reasoning technical report." arXiv preprint arXiv:2504.21318 (2025).
>
> [2] Tu, Songjun, et al. "Learning When to Think: Shaping Adaptive Reasoning in R1-Style Models via Multi-Stage RL." arXiv preprint arXiv:2505.10832 (2025).
>
>
> **Q3: What distinguishes ThinkPrune from the existing length-penalty-based methods?**
>
> We appreciate the reviewer’s observation and agree that our method can be viewed within the broader family of length-penalty approaches. Below we clarify the key distinctions and advantages of our design. Particularly, existing length penalty methods can be formulated as:
> $$
> R(x,y) = C(y) + \lambda S(y),
> $$
> where $x$ is the input and $y$ is the response. $C(y)$ is the correctness reward, $S(y)$ is the length penalty, and $\lambda$ is the weight for the penalty. For Kimi-1.5, $S(y)$ is defined as:
> $$
> S(y) = 0.5 - \frac{L(y)-L_{\text{min}}}{L_{\text{max}}-L_{\text{min}}}, \text{if y is correct}\\
> S(y) = \text{min}[0, 0.5 - \frac{L(y)-L_{\text{min}}}{L_{\text{max}}-L_{\text{min}}}], \text{if y is wrong}
> $$
> To reduce the length, The length-penalty methods require careful manual design $S(y)$ and tuning of $S$. In contrast, our approach introduces **no extra hyperparameters**. Also, it is hard to control the length of the LLM via length penalty methods, since it is hard to find the right $\lambda$ that corresponds to the target length (say if the target length is around 1,000, it is hard to find what values the $\lambda$ should be set to). In contrast, our method allows **direct, explicit control** over the generation length during training by setting a maximum token limit. This cutoff value is intuitive and easy to adjust for practical deployment scenarios.
>
> We also include direct comparisons with models trained using length penalty in EfficientReason [1] and Kimi-1.5 [2] (See **Q1** for more detailed results), where we empirically observe our method brings better performance. In summary, our method brings a **simpler, parameter-free**, and **more interpretable** alternative to length-penalty methods, while delivering **stronger empirical performance** across multiple models and datasets.
>
>
>
> [1] Arora, Daman, and Andrea Zanette. "Training language models to reason efficiently." arXiv preprint arXiv:2502.04463 (2025).
>
> [2] Team, Kimi, et al. "Kimi k1. 5: Scaling reinforcement learning with llms." arXiv preprint arXiv:2501.12599 (2025).

---

### Decision · Action_Editor_vXCk · 2025-12-16

**Recommendation:** Accept with minor revision

**Additional Comments:**

Please incorporate the revisions and clarifications suggested by the reviewers, as addressed in the rebuttal phase, into the final version of the paper.

**Audience:**

Yes

**Audience Explanation:**

Reducing reasoning length is an active research topic, and the paper’s simple and effective approach is relevant to TMLR’s audience.

**Claims And Evidence:**

Yes

**Claims Explanation:**

The authors propose an RL objective to mitigate the overthinking problem in large reasoning models. The effectiveness of the approach is evaluated across multiple models, and the authors further strengthen their claims by introducing additional comparisons with alternative length regularization methods in the rebuttal. They also provide clarifications on the experimental setup, which improves clarity and makes the empirical evidence more convincing.